# Capacitive-Type Pressure Sensor for Classification of the Activities of Daily Living

**Ji Su Park** [1], **Sang-Mo Koo** [2,*] **and Choong Hyun Kim** [1,*]

1   Center for Bionic, Korea Institute of Science and Technology, Seoul 02792, Republic of Korea
2   Electronic Materials Engineering, Kwangwoon University, Seoul 01890, Republic of Korea
*   Correspondence: smkoo@kw.ac.kr (S.-M.K.); chkim@kist.re.kr (C.H.K.)

**Abstract:** In order to operate a gait rehabilitation device, it is necessary to accurately classify the states appearing in activities of daily living (ADLs). In the case of force sensing resistors (FSRs), which are often used as pressure sensors in gait analysis, it is desirable to replace them with other sensors because of their low durability. In the present study, capacitive-type pressure sensors, as an alternative to FSRs, were developed, and their performance was evaluated. In addition, the timed up and go test was performed to measure the ground reaction force in healthy individuals, and a machine learning technique was applied to the calculated biosignal parameters for the classification of five types of ADLs. The performance evaluation results showed that a sensor with thermoplastic polyurethane (substrate and dielectric layer material) and multiwall carbon nanotubes (conductive layer) has sufficient sensitivity and durability for use as a gait analysis pressure sensor. Moreover, when an overlapping filter was applied to the four-layer long short-term memory (LSTM) or the five-layer LSTM model developed for motion classification, the precision was greater or equal to 95%, and unstable errors did not occur. Therefore, when the pressure sensor and ADLs classification algorithm developed in this study are applied, it is expected that motion classification can be completed within a time range that does not affect the control of the gait rehabilitation device.

**Keywords:** capacitive-type pressure sensor; force sensing resistors (FSRs); activities of daily living (ADLs); ground reaction force (GRF); center of pressure (COP); insole device; machine learning; long short-term memory (LSTM)

## 1. Introduction

Activities of daily living (ADLs) refer to bodily movements performed in human daily life, including sitting, standing, walking, climbing stairs, and lifting objects. If the motor function of the human body becomes impaired, then an ADL performance becomes difficult, which can degrade quality of life owing to the reduced levels of activities and shrinking life boundaries [1,2]. Therefore, the early detection of motor impairment and appropriate treatment of patients with gait disturbance are crucial for improving the quality of daily life through enhanced motor ability [3].

The methods for the detection and assessment of human ADLs reported in previous studies have included the direct inspection method [4], use of fixed sensors [5], and use of wearable sensors [6]. Wearable sensors, such as inertial measurement unit (IMU) and miniature pressure sensor, can be used to measure acceleration of the human body [7–10] and the ground reaction force (GRF) generated during gait [10–17].

Studies using IMUs have demonstrated high motion classification sensitivity and ease of implementation because commercial mobile phones or smartwatches can be used or IMUs can be handled directly [8]. However, in studies using IMU sensors, changing the number of used sensors and their attachment position requires the development of a new detection algorithm that can alter the motion classification sensitivity [9]. Moreover, another major disadvantage is that IMU devices should be attached to the body, which can

be burdensome when the study population consists of the elderly or patients with health problems [10].

In contrast, using a pressure sensor attached to an insole device has the advantages of allowing for the standardization of GRF measurement methods and avoiding burden in its application, unlike using IMUs, as the insole is placed inside a shoe. Accordingly, it has been widely used in studies on gait phase classification [10–12], abnormal gait analysis [13], and ADL classification [14]. Force sensing resistors (FSRs) that are thin and capable of low-voltage operation are often used as pressure sensors in insole devices [15]. The load in FSRs is measured based on resistance changes. Overall, FSRs are easy to implement, but they require an amplification circuit because of a small signal size, can be easily damaged by external load [16], and have hysteresis 10 times as large as that of the load cell [17].

To overcome the aforementioned disadvantages of FSRs, many researchers have conducted studies on the development of capacitive-type pressure sensors. In a capacitive-type pressure sensor, the load magnitude is evaluated by measuring the change in permittivity resulting from the dielectric layer thinning caused by load application on the sensor [18]. Pressure sensor implementation based on this method has the advantages of increasing the lifespan of sensors and reducing the production cost [17,19]. In addition, such a sensor can be designed in any desired shape using 3D printing. Moreover, capacitive-type pressure sensors can be used as pressure sensors for measuring GRF owing to lower linear errors and higher sensitivity than those of FSRs [20]. In this regard, insoles with capacitive-type pressure sensors have already been produced, and their performance was tested [21–23].

In a previous study by the authors [24], FSRs were used to measure the ground reaction force, and ADLs were classified into standing and walking. However, for reliable gait rehabilitation, it is necessary to classify the ADLs states in more detail. There is also a need to replace low-durability FSRs with capacitive-type sensors. In this study, a capacitive-type pressure sensor was manufactured and applied to the insole device. In addition, we developed a new algorithm with the ability to classify sitting, sit-to-stand transition, and turning motions, as well as the existing standing and walking motions.

## 2. Methods

### 2.1. Sensor Development

As shown in Figure 1a, the capacitive-type pressure sensor consisted of four layers: a bottommost substrate, two electrodes stacked on its top, and a dielectric layer in between the two electrodes. Polydimethylsiloxane (PDMS) and thermoplastic polyurethane (TPU) were used as the materials for the substrate and dielectric layers, respectively. PDMS has excellent flexibility and is easy to fabricate [25], whereas TPU has higher permittivity than PDMS [26]. Multiwall carbon nanotubes (MWCNTs) and silver nanoparticles (AgNPs) were used as conductive powder materials for electrode fabrication. These materials have excellent electrical conductivity and a small particle size, allowing them to be uniformly dispersed in a polymer material.

For the sensor manufactured using PDMS (hereinafter, PDMS sensor), the substrate layer was fabricated by mixing PDMS (Dow Inc., Michan, MI, USA) and 10 wt% N-hexane (Dow Inc., Michan, MI, USA). A mold was used to fabricate the substrate layer as follows: PDMS in a liquid state was mixed with N-hexane, after which the mixture was poured into a circular mold (20 mm in diameter) and hardened in a 50 °C oven for 30 min. For the electrode layers, 50 wt% of MWCNTs (CM-100, Hanwha, Incheon, Republic of Korea) or AgNPs (APS 0.7–1.3 μm, Alfa Aesar, MA, USA) was uniformly dispersed on the same material as the substrate layer, after which the mixture was molded in a round shape (15 mm in diameter) on top of the hardened substrate layer and hardened in a 50 °C oven for 30 min. The dielectric layer was molded and hardened on top of the electrode layer in the same way as the substrate layer, whereas the final electrode layer was molded and hardened on top of the dielectric layer in the same way as the first electrode layer.

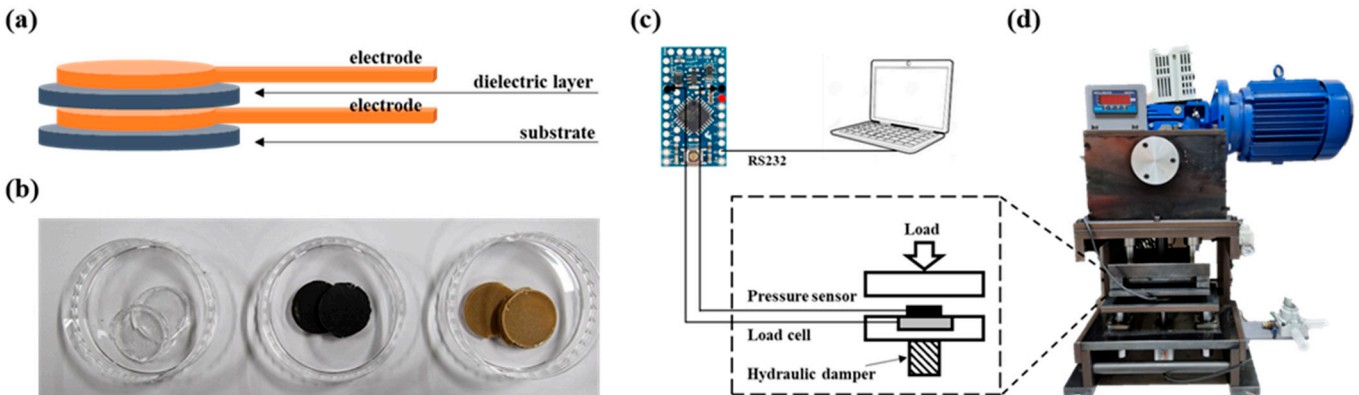

**Figure 1.** Structure of the developed capacitive sensor and the sensor test setup: (**a**) schematic illustration of the capacitive sensor; (**b**) photo image of the manufactured sensor layers (from left: substrate/dielectric layer with PDMS and TPU; electric layer with MWCNTs; electric layer with AgNPs); (**c**) data acquisition system using Arduino; (**d**) sensor durability test rig.

For the sensor manufactured using TPU (hereinafter, TPU sensor), hardened TPU was dissolved with a solvent, unlike PDMS. The solvent used was prepared by mixing N,N-dimethylformamide (Deajung Co. Siheung-si, Republic of Korea) and tetrahydrofuran (Deajung Co. Siheung-si, Republic of Korea) at a 1:4 ratio by volume. The TPU and this solvent were mixed at a 1:2 ratio and dissolved for 48 h by shaking in an orbital shaker at 60 rpm to prepare the TPU mixture. The TPU sensor was manufactured by pouring the TPU mixture in a mold and by hardening in a 40 °C oven for 30 min.

For both the PDMS and TPU sensors, the thickness of the substrate and dielectric layers was approximately 300 μm, and the final sensor thickness was approximately 1 mm. Moreover, the outer diameter of the sensor and the diameter of the sensing area (i.e., electrodes) were fabricated to be approximately 20 and 15 mm, respectively. The manufacturing of the pressure sensor was completed by using a conductive paste to connect the DC power supply wires to the two electrodes. The detailed properties of the dielectric layer are described in Appendix A.

The performance of the manufactured pressure sensors was evaluated based on the following three parameters: sensitivity, hysteresis, and durability. The sensor sensitivity was evaluated by comparing the load value measured when the weights of 20 N (pressure: 110 kPa), 40 N (pressure: 220 kPa), 60 N (pressure: 330 kPa), and 80 N (pressure: 440 kPa) were placed on top of the sensor. For the evaluation of hysteresis and durability, a durability test rig was used with the load repeatedly applied to the sensor at a 2 Hz cycle. For hysteresis measurement, a repeated load was applied five times. For the durability test, 80 N of static load was applied to the pressure sensor, and the maximum capacitance value was measured. This process was repeated 100,000 times to observe a decreasing trend in capacitance.

As described above, a total of four different sensors with different combinations of dielectric layers and electrode materials were manufactured, and the most suitable sensor was selected based on the evaluation of the performance of each pressure sensor.

### 2.2. Hardware Description

Figure 1 shows the developed capacitive-type sensor and the performance test setup. Figure 1a shows the single-layer structure of the sensor, and Figure 1b shows actual photo images of the fabricated substrate, dielectric, and electrode layers. Figure 1c depicts the measurement procedure of a change in the capacitance of the sensor using Arduino (Arduino S.r.l., Scarmagno, Italy). In this procedure, the time for the voltage at both ends of the sensor to reach 63.2% of the maximum voltage was measured, namely, the RC time constant representing the capacitor charging time. Figure 1d shows the durability test rig that repeatedly applied the load on the sensor using an electric motor and a cam shaft.

The rig was set up to control the magnitude and cycle of the applied load. The magnitude of the applied load was measured using a load cell (CWW11-K100, DACELL, Chungju-si, Republic of Korea) and relayed to Arduino, as shown in Figure 1c. The capacitance and load data measured by the pressure sensor and the load cell relayed to Arduino were transmitted to a PC via wired communication (RS-232) with a sampling rate of 1 kHz to be saved. After analyzing the data, two types of data were compared to evaluate the load measurement performance of the capacitive-type sensors developed in the present study.

Figure 2 shows the insole system used in the present study, which consisted of (1) pressure sensors; (2) an insole where the pressure sensors were attached; (3) PCB for GRF data processing. This insole system was the same system that the authors used in their previous studies on fall detection and gait analysis [24,27]. Figure 2a,b show the insole connected to the data processing PCB and capacitive-type pressure sensors; placement of the insoles inside the shoes; and an illustration of the GRF data processing. The GRF data measured using the insole system were collected via Bluetooth communication with a sampling rate of 200 Hz and saved on a PC to be analyzed.

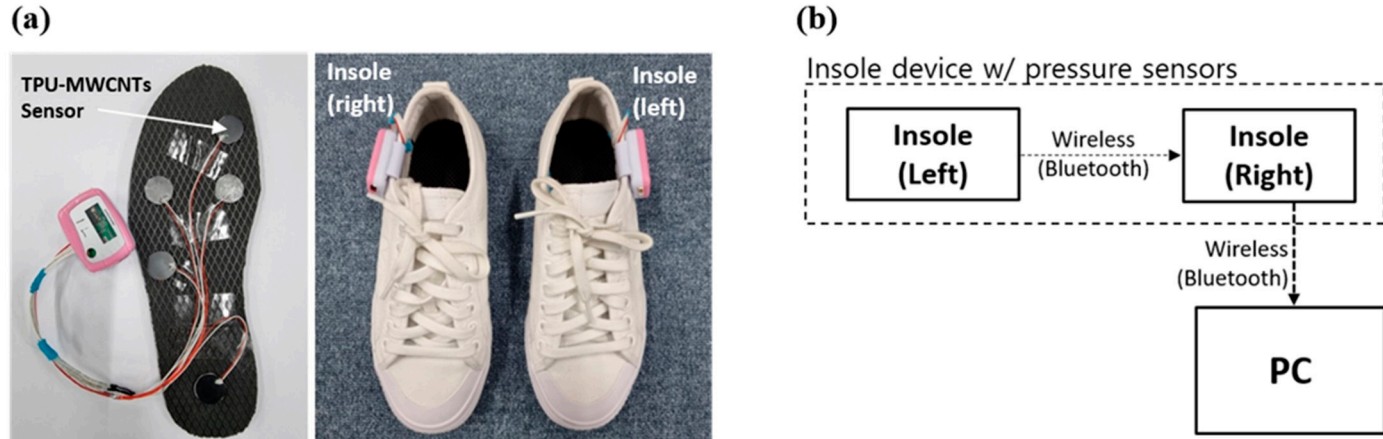

**Figure 2.** Insole system for ADL classification: (**a**) using capacitive-type pressure sensors; (**b**) composition of the data acquisition system.

Finally, the capacitive-type pressure sensors developed in the present study and conventional FSRs were applied as pressure sensors in the insole system for use in a gait experiment. The GRF data acquired through the experiment were analyzed, and the performance was compared.

### 2.3. Participants and Experimental Procedures

A single participant (male, height: 1.73 m, weight: 56 kg, and age: 26) with no history of gait disturbance took part in the gait experiment. The experiment was carried out using the timed up and go (TUG) test method [28]. As shown in Figure 3, the TUG test consists of the participant sitting in a chair, standing up from the chair, walking forward for 3 m, turning around at the turning point, walking back to the chair, and sitting at the chair. This process involves five different motions: sitting, sit-to-stand transition, standing, walking, and turning.

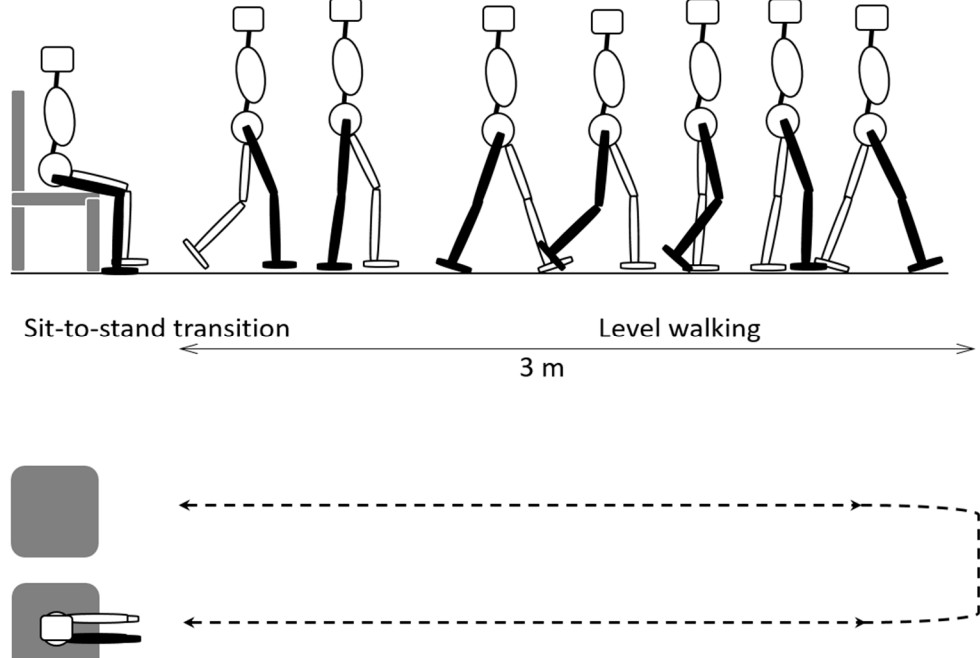

**Figure 3.** TUG test with five types of ADLs: sitting, sit-to-stand transition, standing, walking, and turning.

These five motions were simultaneously measured using a motion capture system (Osprey Motion Analysis Corporation, Santa Rosa, CA, USA). To compare the GRF data acquired with the insole system and the motion data measured using the motion capture system, a third device was used to generate synchronization signals to synchronize the two sets of data [24]. The synchronized GRF data were used to classify the motions of the participants, and the results were compared with the motion detection results obtained with the motion capture system to evaluate whether the five motions from the TUG test could be classified with the insole system.

The TUG test was repeated 30 times, but the experimental data from the first five tests were excluded, because subjects typically move awkwardly at the beginning when instructed to walk forward. Accordingly, the motion data from the remaining 25 tests were used in the analysis.

The experimental protocol was approved by the Institutional Review Board (IRB) at the Korea Institute of Science and Technology (KIST). The participant provided written informed consent for the study prior to participation.

*2.4. Biosignal Parameters*

The GRF data acquired with the insole system were used to calculate the parameters listed in Table 1. These parameters were also used by the authors in a previous study on the classification of the standing and walking motions in healthy individuals and patients [24]. In Equation (8), L represents the pelvic width calculated based on the participant's height. In Equation (9), the time interval was set to 5 ms. In Equation (10), the window size for calculating the waveform length was 200 ms (Table 1).

**Table 1.** Calculated biosignal parameters using ground reaction force data.

| Parameter | Equation | | Description |
|---|---|---|---|
| $F_i$ | $F = 19.734 \times Capacitance - 7.653$ | (1) | Ground reaction force (GRF) |
| $COP_X$, $COP_{RX}$, $COP_{LX}$ | $COP_X = \sum_{i=1}^{10}(F_i \times x_i) / \sum_{i=1}^{10} F_i$ | (2) | Center of pressure (COP), x-axis |
| | $COP_{RX} = \sum_{i=6}^{10}(F_i \times x_i) / \sum_{i=6}^{10} F_i$ | (3) | |
| | $COP_{LX} = \sum_{i=1}^{5}(F_i \times x_i) / \sum_{i=1}^{5} F_i$ | (4) | |
| $COP_Y$, $COP_{RY}$, $COP_{LY}$ | $COP_Y = \sum_{i=1}^{10}(F_i \times y_i) / \sum_{i=1}^{10} F_i$ | (5) | Center of pressure (COP), y-axis |
| | $COP_{RY} = \sum_{i=6}^{10}(F_i \times y_i) / \sum_{i=6}^{10} F_i$ | (6) | |
| | $COP_{LY} = \sum_{i=1}^{5}(F_i \times y_i) / \sum_{i=1}^{5} F_i$ | (7) | |
| $COP_{gradient}$ | $\arctan((COP_{RY} - COP_{LY}) / L)$ | (8) | Gradient of COP |
| $\dot{COP}$ | $\dot{COP} = \dfrac{d}{dt} COP_{gradient}$ | (9) | Angular velocity of the gradient of the COP |
| $COP_W$ | $COP_W = \sum_{i=a}^{window\ size} COP_{gradient}$ | (10) | Waveform length of the gradient of the COP |

### 2.5. Classification of ADLs

The present study attempted to use a machine learning technique to classify five types of ADLs described in the TUG test section, whereas the aforementioned biosignal parameters were calculated and used as input data.

The machine learning technique used was long short-term memory (LSTM), which is a type of recurrent neural network (RNN). Figure 4 shows the LSTM model used in the present study. This model carried out learning using a fully connected layer consisting of 50 nodes and LSTM consisting of 30 nodes with the input of 19 biosignals (10 GRF data, 6 COP data, $COP_{gradient}$, $\dot{COP}$, and $COP_w$).

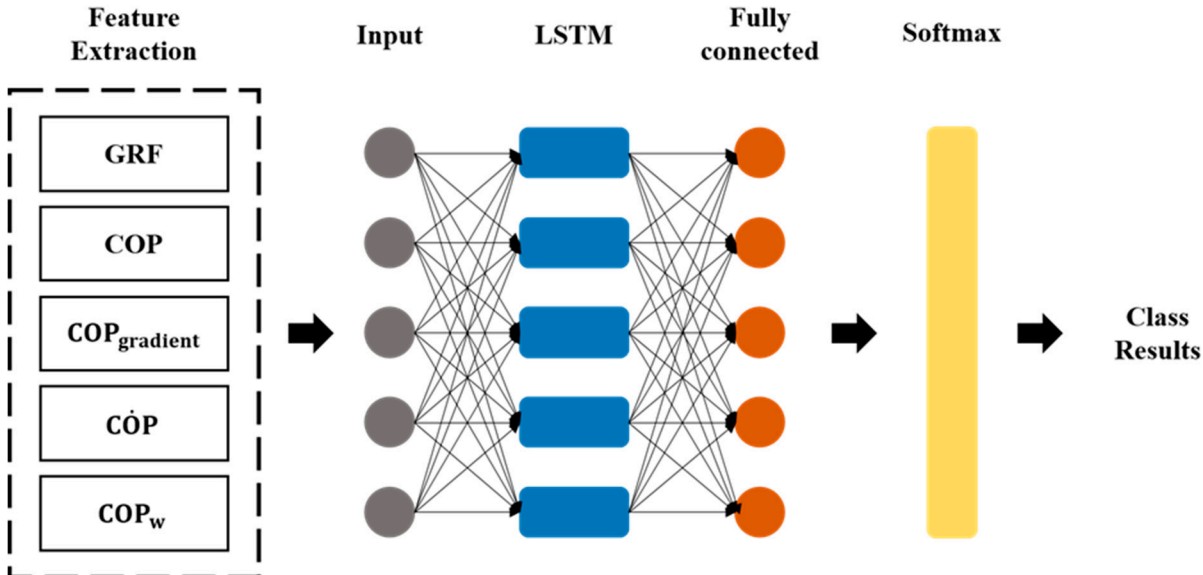

**Figure 4.** Structure of the LSTM model used to classify the ADLs.

Before proceeding with machine learning, the 19 biosignals were subjected to a Butterworth low-pass filter with a frequency cut of 20 Hz to remove noise. In addition, in order to exclude the influence of the magnitude of the absolute value of each biosignal, a normalization process of dividing the maximum value of each biosignal into a size between 0 and 1 was performed.

When applying the LSTM technique, the feedback of data from the section 200 ms earlier and the precision of each model were calculated as the number of LSTM layers increased from 1 to 5. Moreover, an artificial neural network (ANN) algorithm designed to have a four-layer structure with 30 nodes was developed to evaluate the precision of the ADL classification. The result was compared with the precision of the LSTM algorithm.

The nodes of the ANN layer were composed of Relu, and both LSTM and ANN models were created using the Keras library. For the model training, the results were compared after 1000 epochs were performed.

To prevent overfitting of the ADL classification model, $4 \times 4$ fold cross-validation was performed, and the experimental data were divided into 70% training data and 30% test data. In other words, of the 25 experimental data sets used in the analysis, the numbers of training and test data sets were 18 and 7, respectively. Of the training data, three data sets were used for model validation. This cross-validation was performed four times by randomly dividing the entire data in a different manner.

Moreover, the aforementioned LSTM algorithm was applied to the GRF data acquired using FSRs and capacitive-type pressure sensors to calculate and compare the precision of the ADL classification.

## 3. Experimental Results

### 3.1. Sensor Selection

The sensors developed in the present study were divided into four types by classifying them based on the combinations of materials used (dielectric layer material–electrode material): PDMS-MWCNTs, PDMS-AgNPs, TPU-MWCNTs, and TPU-AgNPs. The sensor performance was evaluated based on the sensitivity, hysteresis, and durability. Figure 5 and Table 2 show the results.

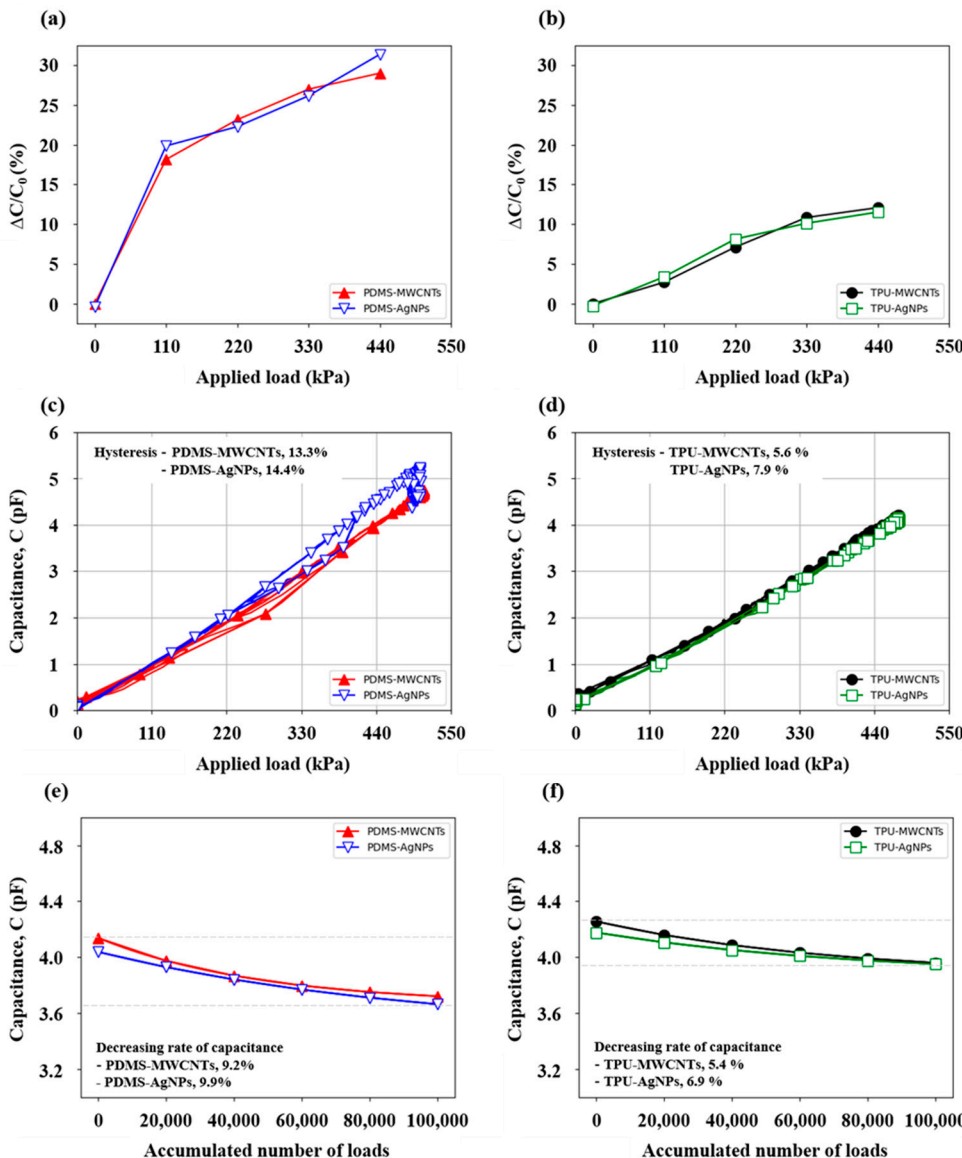

**Figure 5.** Characteristics of the developed capacitive-type sensors: (**a**,**b**) capacitance changes of the PDMS sensors and the TPU sensors; (**c**,**d**) hysteresis curves of the PDMS sensors and the TPU sensors; (**e**,**f**) durability test results of the PDMS sensors and the TPU sensors.

**Table 2.** Characterization of the capacitive-type sensors.

| Sensor Type | Sensitivity (pF/kPa) | Hysteresis (%) | Durability |
|---|---|---|---|
| PDMS-WMCNT | 0.033 | 13.3 | 9.2% decreased |
| PDMS-AgNPs | 0.035 | 13.4 | 9.9% decreased |
| TPU-MWCNT | 0.028 | 5.6 | 5.4% decreased |
| TPU-AgNPs | 0.025 | 7.9 | 6.9% decreased |

However, a linear regression analysis could not be applied to the sensor sensitivity data shown in Figure 5a,b, because the PDMS sensor had different rates of change in the capacitance between the low- and high-load regions [29]. Figure 5a,b show the capacitance at no applied load ($C_0$) and the rate of increase in capacitance with increasing applied load ($\Delta C$) for the PDMS and TPU sensors. Consequently, the capacitance data measured in the load region exceeding 110 kPa (20 N load) were used to perform the linear regression

analysis. The obtained sensitivities of the PDMS-MWCNTs, PDMS-AgNPs, TPU-MWCNTs, and TPU-AgNPs sensors were 0.033, 0.035, 0.028, and 0.025 pF/kPa, respectively, indicating that the PDMS sensors had a 28.3% higher sensitivity, on average, than the TPU sensors.

Figure 5c,d show the hysteresis curves of the four sensors developed in the present study. The hysteresis curves were derived by examining the changes in the capacitance in the same pressure interval measured under loading/unloading conditions when the load was repeatedly applied five times at a 2 Hz cycle. The results show that the maximum hysteresis of the pressure sensors was 13.3% at 275 kPa, 14.4% at 375 kPa, 5.6% at 250 kPa, and 7.8% at 275 kPa for the PDMS-MWCNTs, PDMS-AgNPs, TPU-MWCNTs, and TPU-AgNPs sensors, respectively. The hysteresis of generally used FSRs is approximately 10%, whereas the hysteresis of the PDMS and TPU sensors developed in the present study was higher by 3.58% and lower by 3.3%, on average, than 10%, respectively.

The durability test results for the pressure sensors are shown in Figure 5e,f. As can be observed, after 100,000 trials of 80 N static application, the PDMS-MWCNTs, PDMS-AgNPs, TPU-MWCNTs, and TPU-AgNPs sensors showed to be 9.2%, 9.9%, 5.4%, and 6.9% relative to the baseline capacitance, respectively. In other words, the rate of decrease in the capacitance was higher by an average of 3.4% in the PDMS sensors than in the TPU sensors. When the same dielectric material was used, the rate of decrease in the capacitance was greater by an average of 1.1% in the AgNP electrodes than in the MWCNT electrodes.

In summary, the sensor sensitivity was higher in the PDMS sensors than in the TPU sensors, but the PDMS sensor showed an inconsistent sensitivity at different magnitudes of load. Furthermore, the PDMS sensors had lower durability than the TPU sensors. Meanwhile, there was no significant difference in the performance between the MWCNTs and AgNPs that were used as electrode materials, but the MWCNTs had a significantly lower cost. In this regard, the TPU-MWCNTs sensor was selected as the pressure sensor for the TUG test and was used in manufacturing the insole system explained in Figure 2a.

### 3.2. Biosignal Parameters

The selected TPU-MWCNT sensor was used to derive Equation (1) for measuring the first parameter in Table 1 (GRF). The results are shown in Figure 6. Subsequently, this equation was used to calculate the GRF, and the results were used in the analysis.

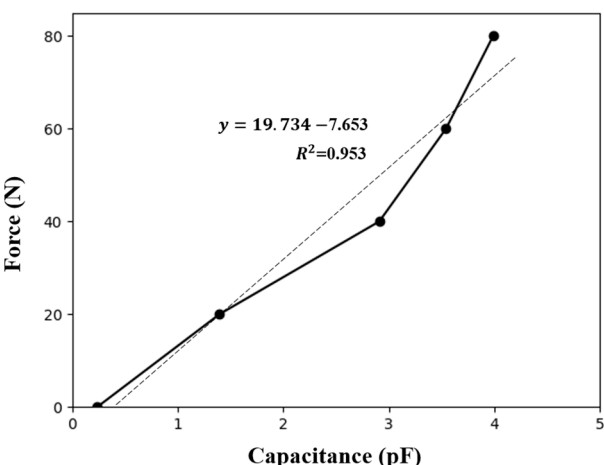

**Figure 6.** Curve fitting of the capacitance–force relationship for the TPU-MWCNT sensor.

Figure 7 shows the biosignal parameters calculated using the GRF data acquired from the TUG test. The numbers on top of Figure 7 represent the five experimental motions classified using the data measured by the motion caption device: (1) sitting, (2) sit-to-stand transition, (3) standing, (4) walking, and (5) turning. With respect to the biosignal parameters, the $COP_X$, $COP_{gradient}$, $\dot{COP}$, and $COP_w$ data showed a similar change in the graphs between the sitting, sit-to-stand transition, and standing motions and the walking

and turning motions, complicating their classification. In contrast, the comparison of the GRF and $COP_Y$ data showed that the data for the sit-to-stand transition motion can be classified because this type of motion differs from the other motions.

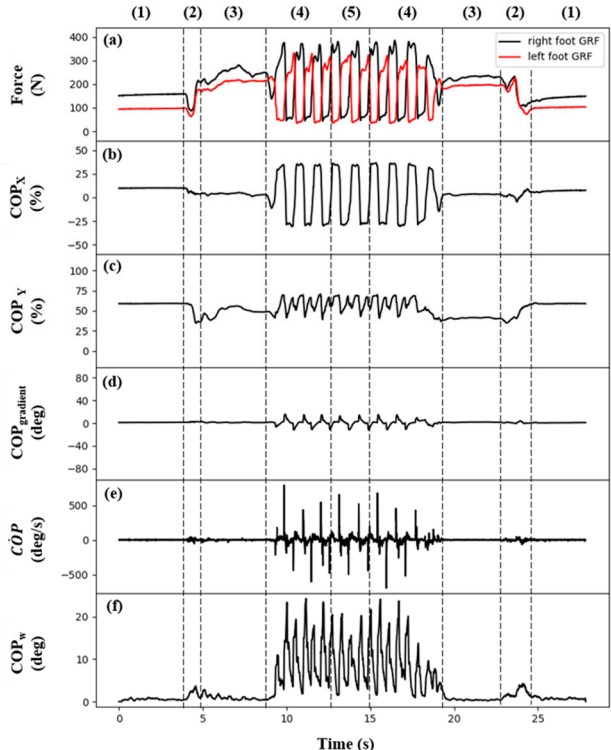

**Figure 7.** Typical test results measured in the TUG test using the smart insole: (**a**) GRF; (**b**) $COP_X$; (**c**) $COP_Y$; (**d**) $COP_{gradient}$; (**e**) $\dot{COP}$; (**f**) $COP_w$. Each number label corresponds to the ADLs as follows: (1) sitting, (2) sit-to-stand transition, (3) standing, (4) walking, and (5) turning.

### 3.3. Classification of ADLs

Five types of LSTM models and one type of ANN model were applied to the GRF data acquired using the capacitive-type pressure sensors for the classification of ADLs in the TUG test. Figure 8 shows the precision of the ADL classification based on the obtained results. The LSTM models showed an average classification precision of 99.24%, 94.99%, 99.28%, 97.50%, and 87.60%, and the ANN model showed an average classification precision of 99.69%, 93.59%, 98.56%, 96.85%, and 83.56% for the sitting, sit-to-stand transition, standing, walking, and turning motions, respectively. When the FSRs were used, the four-layer LSTM model showed an average classification precision of 99.66%, 98.33%, 99.44%, 99.61%, and 91.46%, and the five-layer LSTM model showed an average classification precision of 99.98%, 98.10%, 99.14%, 99.32%, and 98.43% for the sitting, sit-to-stand transition, standing, walking, and turning motions, respectively.

Existing studies have reported that a motion classification algorithm for treatment or rehabilitation should have a precision exceeding 90% [24]. The precision of the LSTM models developed in the present study exceeded 90% for the sitting, sit-to-stand transition, standing, and walking motions, but the classification precision for the turning motion was lower than 90%. As shown in Figure 7, the GRF and $COP_Y$ data were very similar to each other for the walking and turning motions, and as a result, it is difficult to differentiate between them, which could result in the low classification precision for the turning motion. The four-layer and five-layer LSTM models showed turning motion detection precisions of 95.09% and 99.93% (both exceeding 90%), respectively. The five-layer LSTM model showed a very high motion detection precision, with a probability of motion detection error of lower than 1% for all five motions.

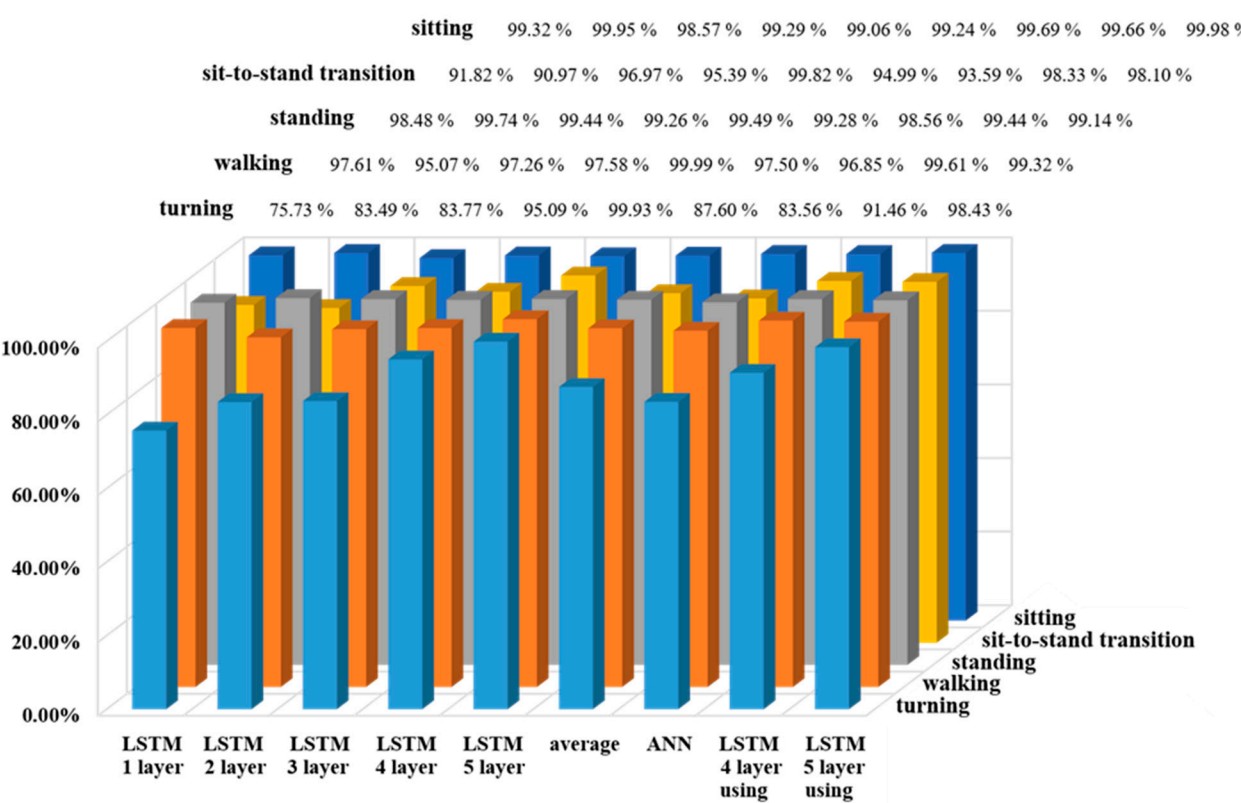

**Figure 8.** Comparison of the precision data for the ADL classification using the LSTM and ANN methods.

As shown in Figure 8, even when the FSRs were used, the classification precision of all five motions exceeded 90% when the four-layer and five-layer LSTM models were applied. In particular, the turning motion detection precisions were 91.46% and 98.43%, respectively. Although these results were 3.36% and 1.50% lower than the corresponding results for the TPU-MWCNT sensor developed in the present study (95.90% and 99.93%, respectively), they still represent an excellent precision.

Figure 9 shows the confusion matrices of the motion classification precision using four different methods (four-layer and five-layer LSTM models applied to TPU-MWCNTs and FSRs). The y-axis of the confusion matrix shows the actual state of the ADLs detected by the motion capture system, and the x-axis shows the state predicted by the detection algorithm. The numbers inside the dark blue boxes located diagonally represent the precision values. Figure 9c shows the four-layer LSTM model applied to the FSRs, where the rate of the algorithm accurately classifying the turning motion (predicted label) relative to the actual turning motion (number label 4, true label) was 0.9146 (91.46%). This value was the lowest among the ADL classification precisions listed in Figure 9. Meanwhile, in Figure 9a, which shows the four-layer LSTM model applied to the TPU-MWCNT sensor, the rate of the algorithm accurately detecting the turning motion was 0.9509 (95.09%). In contrast, applying the five-layer LSTM model resulted in a high classification precision exceeding 98%, regardless of the sensor type.

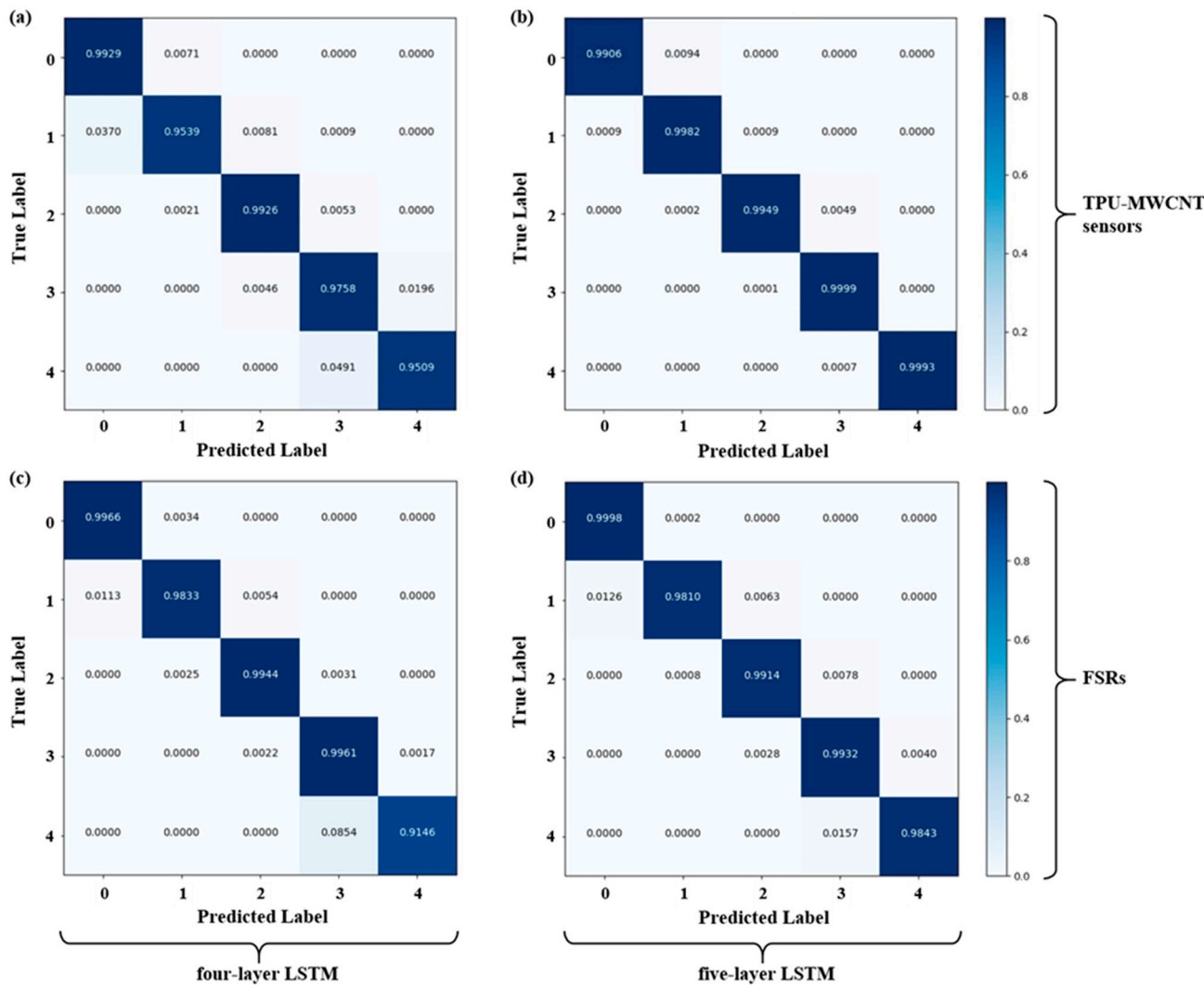

**Figure 9.** Confusion matrices of the classification model: (**a**) four-layer LSTM and TPU-MWCNTs sensor; (**b**) five-layer LSTM and TPU-MWCNTs sensor; (**c**) four-layer LSTM and FSRs; (**d**) five-layer LSTM and FSRs. Each number label corresponds to the ADLs as follows: (0) sitting, (1) sit-to-stand transition, (2) standing, (3) walking, and (4) turning.

Regarding the detection errors for each motion shown in the matrices, there are errors where sitting is mislabeled as sit-to-stand transition, sit-to-stand transition as sitting and standing, standing as sit-to-stand transition and walking, walking as standing and turning, and turning as walking. It should be noted that nonrelated motions are clearly distinguished. For example, in the actual state of walking, the model does not predict sitting, and possible mislabels are only standing and turning. Therefore, the developed algorithm was determined to be reliable.

## 4. Discussion

In the authors' previous study [24], in order to reduce the state classification error caused by shuffling, $COP_w$ was proposed as a new feature to classify the states of standing and walking. Tsukahara et al. conducted a study to distinguish the sit-to-stand transition using the COP change in the gait direction [30]. As shown in Figure 8, using all the algorithms introduced here, it was possible to distinguish the four motions of standing, walking, sitting, and sit-to-stand transition with over 90% accuracy. However, in the case of the turning operation, when the ANN algorithm used in the previous study was applied, the classification accuracy was 83.56%, which did not achieve the required accuracy of 90%.

The reason why the classification accuracy of the turning motion was low can be known by observing the force change appearing in the turning motion using a force plate in Glaister's study [31]. Among the ground reaction forces during walking, the force components in the direction parallel to the ground are braking force, propulsive force, and lateral force. According to the results of Glaister's study, when the motion was changed from walking to turning, the braking force and propulsive force did not show a clear difference among them, but the lateral force was known to show a large change. Unfortunately, the insole device manufactured with thin pressure sensors could not measure the horizontal force (shear force) change on the ground, so turning could not be accurately classified.

To overcome this limitation of the ANN model used in previous studies, we introduced an LSTM model in which the feedback of the biosignals data from the section 200 ms earlier was applied.

Jung et al. classified errors in a detection algorithm for controlling a gait rehabilitation exoskeleton robot into four types: early detection, late detection, unstable error, and combination of the above. These errors were assessed by setting 150 ms as the detection delay time limit that would not significantly affect the robot control [32]. Among these detection errors, unstable errors should be eliminated, because they can cause undesirable vibrations when the process of instantaneously starting and stopping the control device is repeated. As such, an unstable error also occurred in the present study; an overlapping method was used for its elimination. In the overlapping method, the mode value of all motion classification results in the interval 150 ms prior to the detection time point was used as the current motion classification result (Figure 10).

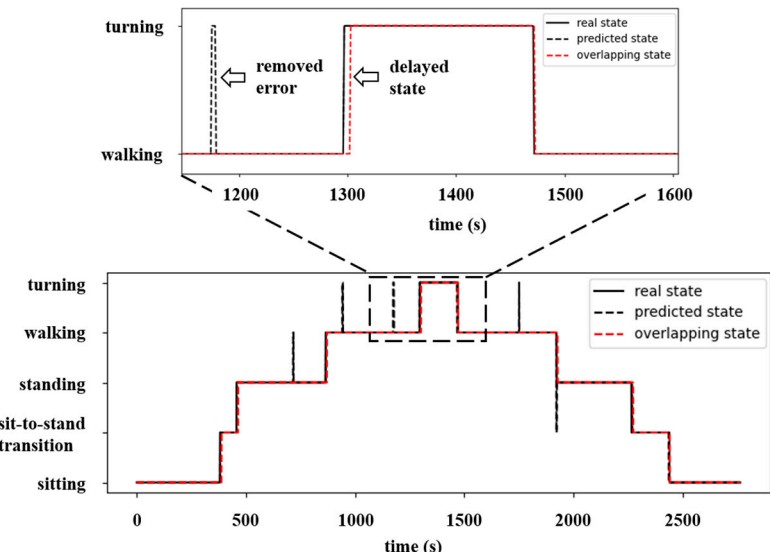

**Figure 10.** Typical result of the classification of ADLs using the four-layer LSTM model.

Figure 10 shows the motion classification results obtained from a series of TUG tests performed using the four-layer LSTM model. Here, the black, solid lines represent the motions detected by the motion capture system; the black, dotted lines represent the motions classified by the LSTM model; and the red, dotted lines represent the motions classified by applying the overlapping method described above. The examination of the classification results obtained using the LSTM model (black, dotted lines) shows that there were no early detection or late detection types of error, whereas an unstable error occurred for walking and standing motions. The results obtained after applying an overlapping filter to these data (red, dotted lines) show that the unstable error was removed, whereas a late detection error with a delay of approximately 70 ms occurred. The magnitude of such a delay was lower than 1/2 of the 150 ms suggested as the limit in the aforementioned study

by Jung et al. Therefore, it was confirmed that the motion classification algorithm with the application of the overlapping method can be used to control exoskeleton robots.

## 5. Conclusions

In the present study, capacitive-type sensors were developed as an alternative to FSRs, which are widely used as pressure sensors for measuring GRF. The newly developed sensors were applied to an insole system, and the insole system was used in TUG tests for the classification of five ADLs: sitting, sit-to-stand transition, standing, walking, and turning.

The experimental results showed that the TPU-MWCNT sensor developed in the present study demonstrates sufficient sensitivity and durability for use as a gait analysis pressure sensor. Moreover, when the GRF data obtained from the TPU-MWCNT sensor were applied to the four-layer or five-layer LSTM model, five motions appearing in the TUG test could be classified with a precision exceeding 95%. Further, when an overlapping filter was applied to the four-layer or five-layer LSTM model developed for use in motion classification, no unstable error occurred, and the detection delay was within the time limit. This indicates that it can be used for the stable control of exoskeleton robots.

As for the biosignals acquired through the TUG tests in the present study, closely related motions, such as walking and turning, were difficult to differentiate. Therefore, future studies should attempt to identify new biosignal parameters for classifying motions among ADLs that are closely related to each other and to conduct motion classification using such parameters.

**Author Contributions:** Conceptualization, J.S.P. and C.H.K.; methodology, J.S.P. and C.H.K.; software, J.S.P.; validation, J.S.P. and C.H.K.; formal analysis, J.S.P. and C.H.K.; investigation, J.S.P., S.-M.K. and C.H.K.; data curation, J.S.P.; writing—original draft preparation, J.S.P.; writing—review and editing, S.-M.K. and C.H.K.; supervision, C.H.K.; project administration, C.H.K.; funding acquisition, C.H.K. All authors have read and agreed to the published version of the manuscript.

**Funding:** This work was supported by the Korea Institute of Science and Technology (KIST) Institutional Program (project nos. 2E31110 and 2E31642).

**Institutional Review Board Statement:** The study was conducted according to the guidelines of the Declaration of Helsinki and approved by the Institutional Review Board of the Korea Institute of Science and Technology (approval no. 2020-001 on 7 April 2020).

**Informed Consent Statement:** Informed consent was obtained from the subject involved in the study.

**Data Availability Statement:** The data presented in this study are available upon request from the corresponding author. The data are not publicly available due to the fact of ethical concerns, because they were obtained in a clinical trial.

**Conflicts of Interest:** The authors declare no conflict of interest.

## Appendix A

The modulus of elasticity of the dielectric layer was measured as shown in Figure A1a. A dielectric layer and a load cell (9047C, Kistler, Switzerland) were placed between the anvil and spindle of a digital micrometer (Mitutoyo, Japan, range 50–70 mm), the displacement and load values were measured while moving the spindle, and the modulus of elasticity was calculated [33]. The resolution of the load cell and digital micrometer was 0.01 N and 1 μm, respectively. In order to uniformly apply a force to the surface of the dielectric layer, round steel plates with a thickness of 1 mm were placed on both sides of the dielectric layer specimen. The capacitance data output from the sensor was collected in Labview (NI9215, National Instruments, Austin, TX, USA) via Arduino. The stress was obtained by dividing the load by the area of the circular steel plate. The modulus of elasticity of the dielectric layer is shown in Figure A1c.

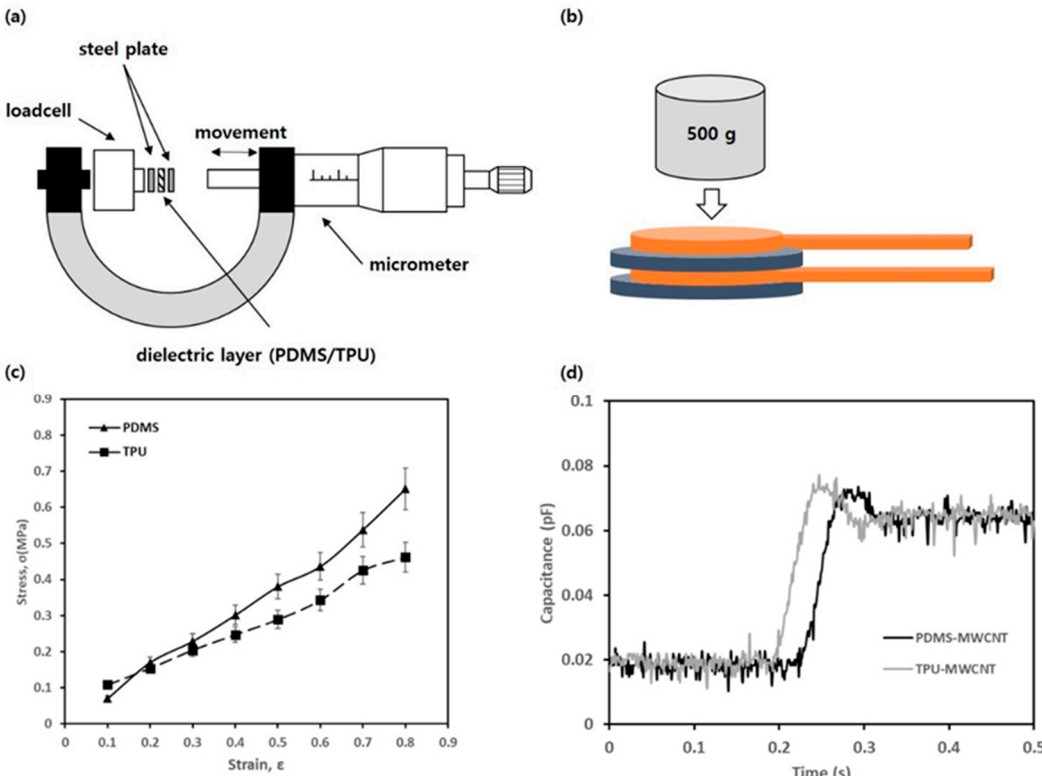

**Figure A1.** Sensor capability of the capacitive-type pressure sensors: (**a**) schematic of the elasticity test setup, where the pressure sensor was attached on the load cell; (**b**) schematic of the response time test setup, where the response of the pressure sensor to the weight drop was 500 g each; (**c**) elasticity of dielectric layers; (**d**) response time of each sensors.

The response time of the sensors was measured as in Figure A1b. The response time was measured as the time required for the capacitance increase rate to reach 10% to 90% after placing a 500 g weight on the dielectric layer specimen. The response time of the dielectric layer is shown in Figure A1d. The response times of the PDMS sensor and the TPU sensor were 35 and 39 ms, respectively. In Yao's study [34], the response time of the pressure sensor was 40 ms, and the response time of the sensors fabricated in this study were 1 to 5 ms faster than this value.

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
