# Peer review of "Capacitive-Type Pressure Sensor for Classification of the Activities of Daily Living"

_2673-8023, doi:10.3390/micro3010004_

Round 1
Reviewer 1 Report
Good work, should be accepted without revision. Existing force sensing resistors are often used as pressure sensors in the gait analysis but with low durability. The author developped the capacitive-type pressure sensors as an alternative to force sensing resistors and evaluated the performance of these sensors by applying to an insole system which were used to classfy five ADLS. This manuscript is well structured and organized, and the experimental data are enough for the verification of viewpoints raised by the authors. Also, the novelty points of the manuscript are highlighted with certain significances, which maybe of interest to the broad readers from such community. Thus, I recommend this paper to be accepted without revision.
Author Response
We want to express our gratitude for your concern and encouragement.
Your positive evaluation of this manuscript gave us the courage to devote ourselves to research. Thank you very much.
Reviewer 2 Report
The authors have developed a daily living activity monitoring system based on capacitive pressure sensors embedded in shoe insoles. This work is of great scientific interest since demonstrated sensors are easy to manufacture and machine learning algorithms applied on generated data allow a decent accuracy in movement classification (some similar movements still require accuracy increase though). Well-established measurement methodology makes presented results trustworthy. I recommend this work to publication after my minor comments and questions are addressed.
1. In sections 3.1 and 3.3 a numerical data on performance of 4 sensor groups is often expressed in text. Please put it into tables for easier data readability.
2. In Figures 5c-f Y axis is labeled as capacitance. Is it delta capacitance?
3. In Figure 5c (hysteresis measurement) there is a notable dip for PDMS-AgNPs sample at 500 kPa. What could be the reason for that?
4. For durability tests up to 80 N force was applied. I would consider applying higher force for durability test knowing that average human weight would create 600-800 N load while walking (and more while jumping). Also, adding higher load would make more datapoints for Figure 6, therefore more precise fitting can be done.
5. In the end of section 3.1 authors mention carbon nanotubes are much cheaper than Ag nanoparticles. Would be nice to have a relative sensor manufacturing cost as an extra metric for 4 sensor groups discussed.
6. In section 3.3 authors discuss ADL classification precision. Decent result was demonstrated with 5-layer LSTM model. 18 data sets were fed into model for training. Can you estimate how much higher classification precision you can obtain if more data sets are used for training? Particularly, for turning movement differentiation from walking movement.
Author Response
Please refer attached file.
